# A Spectroscopic Reflectance-Based Low-Cost Thickness Measurement System for Thin Films: Development and Testing

**DOI:** 10.3390/s23115326

**Published:** 2023-06-04

**Authors:** Néstor Eduardo Sánchez-Arriaga, Divya Tiwari, Windo Hutabarat, Adrian Leyland, Ashutosh Tiwari

**Affiliations:** 1Amy Johnson Building, Department of Automatic Control and Systems Engineering, University of Sheffield, Portobello St., Sheffield S1 3JD, UK; nesanchezarriaga1@sheffield.ac.uk (N.E.S.-A.); w.hutabarat@sheffield.ac.uk (W.H.); a.tiwari@sheffield.ac.uk (A.T.); 2Sir Robert Hadfield Building, Department of Materials Science and Engineering, University of Sheffield, Mappin St., Sheffield S1 3JD, UK; a.leyland@sheffield.ac.uk

**Keywords:** thin film, thickness measurements, reflectometry, sensor, microprocessor, Python, Arduino

## Abstract

The requirement for alternatives in roll-to-roll (R2R) processing to expand thin film inspection in wider substrates at lower costs and reduced dimensions, and the need to enable newer control feedback options for these types of processes, represents an opportunity to explore the applicability of newer reduced-size spectrometers sensors. This paper presents the hardware and software development of a novel low-cost spectroscopic reflectance system using two state-of-the-art sensors for thin film thickness measurements. The parameters to enable the thin film measurements using the proposed system are the light intensity for two LEDs, the microprocessor integration time for both sensors and the distance from the thin film standard to the device light channel slit for reflectance calculations. The proposed system can deliver better-fit errors compared with a HAL/DEUT light source using two methods: curve fitting and interference interval. By enabling the curve fitting method, the lowest root mean squared error (RMSE) obtained for the best combination of components was 0.022 and the lowest normalised mean squared error (MSE) was 0.054. The interference interval method showed an error of 0.09 when comparing the measured with the expected modelled value. The proof of concept in this research work enables the expansion of multi-sensor arrays for thin film thickness measurements and the potential application in moving environments.

## 1. Introduction

In recent years, industrial roll-to-roll (R2R) thin-film deposition processes have seen rapid market growth due to the increasing use of flexible electronics by consumers. This has led to efforts to reduce manufacturing costs and find new ways to digitise these processes. According to a report from Data Bridge Global Market Forecast [1], the R2R market is expected to reach a value of USD 41.55 billion by 2029 with a compound annual growth rate (CAGR) of 21.50% from 2022 to 2029. This trend makes the R2R processes highly attractive for the industry and academia to keep innovating and delivering high-tech and state-of-the-art solutions. One area of focus is in-process inspection, which involves digitising various product parameters, such as coating thickness during the manufacturing process of flexible electronics.

Real-time coating thickness measurements in flexible electronics applications, such as solar cells, micro-electromechanical systems (MEMS), and others, are key for improving the performance of roll-to-roll (R2R) systems [2]. In atmospheric pressure R2R processes, various metrology techniques have been adapted to measure thin film thickness over a long width of the substrate. Ellipsometry has been shown to be capable of performing in-process measurements with 2–10% thickness accuracy, but has presented spatial resolution issues in the central 10 cm of a web span of 30 cm [3,4]. When combined with advanced control methods, interferometry-based techniques such as white light scanning interferometry (WSI) can perform in-line measurements but are limited to single-point inspections and require gaps in the coating to measure thickness [5,6]. Scatterometry-based techniques combined with high-tech cameras have also shown promise in performing in-process measurements, but only measure a single point on the sample [7]. Whilst each of these techniques has advantages and disadvantages in R2R environments [8], this paper shall not provide a detailed examination of those details; instead, the aim is to address the existing limitations in the coverage area for thickness inspection in large substrates.

A commonly used metrology technique for thin film thickness measurements is spectroscopic reflectometry, which is a non-destructive interferometry-based technique for measuring thin film coatings in laboratory environments. It allows for the calculation or estimation of coating thickness without the need for gaps between samples. This technique compares the reflected intensity of uncoated and coated samples using a spectrometer system. By subtracting the measured dark noise of the system, a reflectance curve is generated, which can be used to estimate the coating thickness through several methods such as the interference interval method (IIM) [9,10] or curve fitting method (CFM) [11,12] for thin films (<1 μm) or by Fourier transform for thicker films (>1 μm) [13]. Although this technique is typically used in a laboratory setting for mirror-finish coatings and has limitations such as the 2π phase ambiguity [14], the local minimum [15] and inspection on dark and rough surfaces [16], it has the potential to perform in-process measurements. The work of Grau-Luque, et al. [17], which used normal reflectance for AlOx coating nanolayers on various substrates, such as Si, Cu (In, Ga)Se2 (CIGS) and polyethylene terephthalate (PET), demonstrated the feasibility of reflectance measurements in in-process inspection and highlighted that this is an area with opportunities for further investigation [17]. Although the solution presents a novel approach using reflectometry with machine learning, the scalability of this system still has improvement opportunities to inspect larger surface areas, as the equipment necessary to perform a single-point inspection would require significant space on the manufacturing floor.

Recent advancements in MEMS have led to the miniaturisation of spectrometer (SM) sensors, making them increasingly attractive for industrial applications where the available space to install inspection systems is limited [18]. The miniaturisation has also reduced their cost, improved their specifications, and made them more scalable for in-process inspection in moving environments such as R2R processes. In this study, we have identified two cutting-edge SM sensors that have the potential to be used for spectroscopic reflectometry measurements. Hence, we present a feasibility study in an offline laboratory environment to establish a baseline for future developments. To the best of our knowledge, this is the first time the proposed sensors have been used for thin film measurements in an in situ small package device, as proposed in this research. Although Nemoto presented a work where one of the identified sensors was used for thin film thickness measurements, they were used in a microscope setup, which does not address the process scalability [19]. Additionally, the identified sensors have previously been used for applications such as measuring fruit ripeness, identifying wood defects and other hyperspectral imaging measurements [20,21,22].

Our low-cost system offers superior scalability compared with other options, as it is a compact solution for two SMs in a 13.5 × 30 × 30 cm package. The device can measure the thickness of thin films made from a standard Si:SiO_2_ reference (476.3 nm and 198.7 nm) using two light sources (warm white and cool white) and two methods: the interference interval method (IIM) and the curve fitting method (CFM). By using the IIM, the error was 0.09, and when using the CFM, an RMSE as low as 0.022 was achieved. Additionally, the study reveals that the sensor C12666MA combined with a warm-white light source has the least variation when compared with the combined factors of the C12880MA sensor.

## 2. Theoretical Background

In this research, we employ two methods for characterising the thickness of thin films: CFM and IIM. CFM involves mathematically modelling the reflectance curve of a coated sample and then comparing it with the measured reflectance to determine the film thickness. On the other hand, IIM only uses the measured reflectance to calculate the film thickness without the need for mathematical modelling.

### 2.1. Modelling of Reflectance Curves

To understand spectroscopic reflectometry, it is important to know that it is based on the principles of interferometry, as described in [12,23,24]. In the case of a single semi-transparent substrate such as glass or plastic, with a near-normal angle of incidence (θ_0_ = 0), the Fresnel reflection coefficient (r) can be calculated using the equation $r_{01}=\left(N_{0}-N_{1}\right)/\left(N_{0}+N_{1}\right)$. However, when a thin film is added over the substrate, creating an extra layer, the light propagation changes, as illustrated in Figure 1.

In Figure 1, N_0_ is the refractive index of air, and N_1_ and N_2_ are the refractive indexes of the thin film and the substrate, respectively. The total reflection coefficient is then expressed by the Airy formula [25,26,27]:(1)$$r=\left(r_{01}+r_{12}e^{\left(-i2\phi_{1}\right)}\right)/\left(1+r_{01}r_{12}e^{\left(-i2\phi_{1}\right)}\right),$$
where $\Phi_{1}$ is the phase shift of the light when travelling in the coating expressed as $\Phi_{1}=k^{\prime }dN_{1}\cos\theta_{1}$, where d is the coating thickness, N_1_ is the refractive index of the coating material, θ_1_ is the angle of refraction of the light beam in the coating material, and k’ is the wave number in vacuum. Consequently, k’ = 2π/λ, where λ is the wavelength under study. Assuming low absorption of the coating and that the total reflectance is R = |r|^2^, Equation (1) is rewritten as follows:(2)$$R=\left({r_{01}}^{2}+{r_{12}}^{2}+{2r}_{01}r_{12}\cos2\phi_{1}\right)/\left(1+{r_{01}}^{2}{r_{12}}^{2}+{2r}_{01}r_{12}\cos2\phi_{1}\right),$$
which is used to calculate the reflectance value for a single wavelength (λ); therefore, when analysing a range of wavelengths, a sinusoidal curve is formed [12,24,25]. This formula is applied widely in the field of thin film thickness measurement [12,24] and in other application areas such as the analysis of butterfly wings [25]. By using Equation (2) in a range of wavelengths (λ_i_…λ_n_), a reflectance curve can be modelled across the wavelength range of interest. Consequently, the generated reflectance curve can be compared against reflectance measurements using the CFM (Section 2.3), and the reflectance measurements are performed as explained in Section 2.2.

### 2.2. Reflectance Measurements

Reflectance measurements involve a process in which an *SM*, a microprocessor and a light source are involved. An SM can measure light intensity over a range of wavelengths by assigning specific wavelengths to individual pixels within a line array. The Hamamatsu SM’s C12880MA and C12666MA were identified as being suitable for the research purposes described in this paper. They have 288 and 256 pixels, respectively, where each pixel registers the intensity of a specific wavelength, which can then be combined to create the reflectance spectrum [28]. Hamamatsu provides a formula to determine the specific wavelength of a given pixel as a function of the pixel number:(3)$$\mathrm{Wavelength}\;(\mathrm{nm})=A_{0}+B_{1}x+B_{2}x^{2}+B_{3}x^{3}+B_{4}x^{4}+B_{5}x^{5},$$
where A_0_, B_1_–B_5_ are calibration coefficients provided by Hamamatsu, and x is the pixel number. The relative intensity measurements per pixel depend on the analogue-to-digital converter (ADC) resolution of the microprocessor board that processes the streamed output of the SM. For example, for a 10-bit ADC, the max available counts are 2^10^ − 1 = 1023. Additionally, the integration time of the microprocessor must be adjusted to avoid saturating the maximum intensity counts of the ADC [28].

Three different configurations are required to calculate the reflectance of a coated sample: these are the intensity measurements at (i) no light/dark noise (I_d_), (ii) uncoated sample (I_u_), and, finally, (iii) coated sample (I_c_). The relative intensity measurement values I_d_, I_u_ and I_c_ are then used in the following equation:(4)$$R_{c}=\left(I_{c}-I_{d}/I_{u}-I_{d}\right)(R_{u}),$$
where R_u_ is the standard absolute reflectance of the uncoated sample [11,29,30]. R_u_ is considered as 1 because the absolute reflectance of the uncoated standard is near 1 without causing a major change to the absolute reflectance of the coated sample R_c_ [29]. Once R_c_ is calculated for every pixel, the reflectance at individual bands can be combined into a reflectance curve across the spectrum under test.

The spectral accuracy of the SM critically depends on the bandwidth quality of the light source, which ideally should be a flat white light across the spectrum. In industrial or specialised laboratory deployment, high-quality and wide-bandwidth light sources (UV-VIS-NIR) with a nearly flat white-light spectrum are commonly used, such as xenon and halogen–deuterium (HAL/DEUT).

The resulting reflectance spectrum now can be compared with the mathematically modelled reflectance curve described in Section 2.1. This process is commonly known in the industry as the curve fitting method (CFM) [13].

### 2.3. Curve Fitting Method Deployment

The CFM uses regression methods to compare the modelled with measured reflectance curve. A common method used in curve fitting is to evaluate the root mean squared error (RMSE) of the modelled intensity values per wavelength vs the measured intensity data per pixel [13]. Figure 2 shows an example of curve fitting analysis using two methods: RMSE and normalised mean squared error (MSE_T_) (as proposed by Tompkins [12]).

Both metrics consider an RMSE and MSE_T_ value close to zero as an indicator of good fit quality. Nevertheless, the values can be different as other metrics could account for the standard deviation (σ) of the measured data, such as the MSE_T_ (see Appendix B for the RMSE and MSE_T_ formulae). Some authors suggest performing normalisation using the mean of the measured values [11], and others suggest using the goodness of fit (GOF) [30,31] in which a value close to 1 is an indicator of good fit.

Although the CFM is precise when comparing modelled reflectance curves for specific material thicknesses, it does not measure the coating thickness directly. Another method that can measure the coating thickness directly is the IIM.

### 2.4. The Interference Interval Method

The IIM calculates the thin film thickness by counting the number of waves or interference fringes in an interval/range of wavelengths. Compared with the CFM, this method only requires the measured reflectance curve obtained from Equation (4). As outlined in Shimadzu’s application notes A292 [9] and A614 [32], the film thickness of a single-layer coating is calculated by using the following:(5)$$d=\left(\Delta m/2\sqrt{n^{2}-{sin}^{2}\theta }\right){\left({\lambda_{2}}^{-1}-{\lambda_{1}}^{-1}\right)}^{-1},$$
where Δm is the number of fringes/waves between an interval of wavelengths, n is the known refractive index of the coating, θ is the angle of incidence, λ_2_ is the wavelength of the peak (or valley) located to the left side of the bandwidth under inspection and λ_1_ the corresponding wavelength of the peak (or valley) located to the right side of the bandwidth under inspection. It must be highlighted that the value of λ_1_ must be greater than λ_2_ to guarantee positive results of the calculated thickness value. Figure 3 shows examples of expected SiO_2_ reflectance curves at different thickness values.

Once a reflectance curve is available, the thickness can be calculated using Equation (5), as described in Figure 3. Observe that Figure 3a shows two fringes (Δm = 2) and well-identified valleys (λ_1_ and λ_2_). Figure 3b shows one fringe and two well-identified peaks. Figure 3c shows one fringe and one well-identified valley, but also an estimated valley located at 750 nm (λ_1_). Finally, in Figure 3d the thickness calculation using the IIM is not possible because at least two valleys or two peaks are required to identify a fringe.

Although this method provides a direct thickness measurement, it becomes less accurate for SiO_2_ thicknesses below 400 nm, as the reflectance curve becomes flattened and the peaks/valleys become difficult to identify, as seen in Figure 3d [32]. The reflectance curve changes depending on the refractive index of the coating material; therefore, the IIM limit varies depending on the material nature under inspection.

The CFM is preferred in reflectance setups as it compares the measured reflectance with a mathematical model. It can measure thicknesses below 400 nm with high accuracy, which is not possible with the IIM. A description of a novel thin-film thickness reflectometry system, involving the materials and the hardware and software details to enable the CFM and IIM, is described in Section 3.

## 3. System Development

### 3.1. Hardware Development

The following materials were used in the proposed reflectometry system, as shown in Figure 4: one power supply GPS-4251 GW Instek, one 100 kΩ potentiometer, one LED NSPW315DS Nichia (daylight cool white), one LED NSPL570DS Nichia (warm white), one Avantes reference standard containing two coated samples of 476.3 nm and 198.7 nm, one SM sensor Hamamatsu C12880MA, one SM sensor Hamamatsu C12666MA, one Logic Converter TXS0108E, one Nucleo L432KC microprocessor board, one laptop Dell Latitude 5511, and one IC Dip socket A14-LC-TT.

Power supply and potentiometer: A 3.2 V external power supply in series with a 100 kΩ potentiometer was used as a current limit circuit for the LED under test. This enabled the light intensity adjustment for our experiments.

Reflectometer Assembly: The reflectometer assembly holds one LED and one SM. The LED emits light intensity (green arrow in Figure 4), and then the light is reflected over the coating sample under test (yellow arrow), which is captured by the SM (C12880MA or C12666MA).

Logic converter: The TXS0108E converts the board ST/CLK signal levels from 3.3 V to 5 V to comply with the SM specifications. This ensures a proper voltage level between the SM and microprocessor board (μP board).

Microprocessor Board: The μP board sends start (ST) and clock (CLK) signals to the SM, enabling the video output. Once the SM is enabled, the video signal is sent to the ADC of the μP board to start the video data processing.

The code structure to program the μP board was based on the timing charts of the Hamamatsu SMs and on code found in [33,34,35]. The STM Nucleo L432KC was selected because of its internal OPAMP in the ADC input (PA0) and its oversampling capability to expand up to 16-bit resolution. The Arduino IDE setup was modified to enable the STM board programming according to instructions found in the stm32duino GitHub repository [36] (see Arduino code in Appendix A).

Finally, a laptop DELL Latitude 5511 with NVIDIA GeForce MX250 was used for video data processing and plotting.

Complementing the reflectometer assembly description, 3D-printed parts were designed using Autodesk Fusion 360. These parts enabled the assembly of the SM by incorporating the LEDs for measuring the reflected intensity. Figure 5 provides the details on the device dimensions.

Figure 5a illustrates the dimensions of the 3D-printed device, including the near-normal position of the LED and the light channel slit. To reduce stray light, the sensor holder light channel (Figure 5b) and the internal walls of the LED base (Figure 5c) were coated with antireflective black paint (Rustins BLAB1000). Figure 5d displays the physical device and its method of positioning over the reference standard. The breadboard assembly is shown in Figure 5e, and the circuit can be found in Appendix C. The corresponding .stl files of the 3D-printed parts can be found in Appendix A.

In addition to the reflectometry setup described in Figure 5, the proposed hardware system was tested using another stable light source: Avalight DHS from Avantes and a fibre optic probe FCR-7UVIR200-2. One Thorlabs SMA Fiber Adapter SM05SMA, one 3D-printed probe holder, and one modified version of the sensor holder were used to adapt the fibre optic probe to the reflectometry setup for this set of experiments. The experimental setup with the Avantes light source is described in Section 4.3.

Once the reflectometry setup was properly assembled, the intensity measurements and reflectance calculations were enabled using a Python interface, which is shown in Section 3.2.

### 3.2. Software Development and Measurement Procedure

The open-source software, Python version 3.10, was selected to read the SM intensity measurements per pixel through the USB COM port of the microprocessor board. Figure 6 outlines the measurement procedure, as depicted in Equation (4). Each block represents a Python script and a measurement process step:

Step 1: The intensity calibration script measures the light intensity reflected by the uncoated substrate (Si) per pixel.

Step 2: Once the uncoated sample reference spectrum is selected, then the reference calculation script calculates the average intensity of the last 10 spectrums per pixel. This becomes I_u_ per pixel in Equation (4).

Step 3: The dark noise intensity is measured by turning off the light source first, then performing an average of 10 readings similar to the reference calculation script. Turning off the light source before starting the script is compulsory to capture the dark noise readings accurately. This becomes I_d_ per pixel in Equation (4).

Step 4: The coated sample (Si:SiO_2_) is placed under the 3D-printed device, as shown in Figure 5d, to capture the reflected intensity of the coated sample I_c_. Finally, the reflectance calculation per pixel (R_c_) is performed using Equation (4). Figure 7 shows the outcome of the described steps.

Figure 7a plots the wavelength per pixel vs the reflected intensity, as saved in the CSV files, in each one of the calibration steps. The x-values are the calibrated wavelengths per pixel, as recommended by Hamamatsu in Equation (3). On the other hand, the y-axis values are the relative intensity counts per pixel read from the USB COM port (Iu, Ic and Id). Figure 7b shows the resultant Python reflectance curve; here the y-axis values are the reflectance values per pixel, calculated using Equation (4). The full Python code is available in Appendix A.

Once the intensity and reflectance measurements are enabled, and a reflectance curve is generated, then the fit quality using the CFM and the thickness calculation using the IIM are possible. The following results section shows an experimental plan validating the proposed reflectometry system using both methods, as explained in Section 2.

## 4. Results and Discussions

An experimental plan was created to perform measurements of coating thickness on two samples of Si:SiO_2_ with thicknesses of 476.3 nm and 198.7 nm using the CFM and IIM. Two different spectrometer sensors, two different light sources and two coating samples of Si:SiO_2_ were tested using the CFM and IIM (see Table 1 for specifications). Table 2 shows the results for each combination of factors.

Table 2 shows the results of each one of the 16 experiments. For the CFM data, both sensors presented good RMSE results ≤ 0.05, but the MSE_T_ data presented high-level warnings for both of the sensor combinations. This is because the MSE_T_ accounts for the standard deviation of the measured values. This demonstrates that SENSOR1 presented a higher variation in the measurements compared with SENSOR2; this is because SENSOR1 has a higher sensitivity specification compared with SENSOR2, making it more difficult to calibrate [28]. Again, the results obtained by the CFM show that the combination of SENSOR2 and LED2 presented lower RMSE and MSE_T_ compared with all of the combinations. The CFM results are discussed in Section 4.1.

The results obtained by the IIM gave constant values for SAMPLE1 (476.3 nm) for all combinations of sensors and LEDs. However, because of the limitations of the IIM, the SAMPLE2 (198.7 nm) could not be measured, as discussed in Section 2.4. The IIM results are discussed in Section 4.2.

### 4.1. Curve Fitting Method (Results)

To perform the CFM, a mathematical model of the two samples was created (as discussed in Section 2.1) and the reflectance measurements were performed following the measurement procedure (described in Section 3.2). The following settings were used before analysing the results:Following the literature review, it was found that the fit quality and the choice of performance metric depend on the user’s specific requirements. Therefore, to evaluate the experiments, the RMSE and the MSE_T_ were selected.The choice of quality fit error is dependent on the specific application [11,30]. For this research, a value of RMSE and MSET ≤0.05 was considered a good fit. A value >0.05 is a warning level. Values >0.1 are considered high-level warnings.When measuring the uncoated intensity (Iu) in the calibration step 1 (Section 3.2), the recommendation is to set the highest point of the reflected intensity near 90% of the maximum counts of the 10-bit ADC (ADC resolution = 2^n^ − 1 = 1023 max counts) to avoid SM saturation and data conversion overflow [11]. The highest point of the reflected light captured by the SM was observed, and then the calibration steps were performed accordingly.To expand and repeat the measurements, the uncoated intensity (Iu) was calibrated to 85%, 90% and 95% of the maximum available counts for the ADC.The Arduino integration time variables were fixed to int_Time = 1 (C12880MA) and int_Time = 800 (C12666MA) (34 μs and 0.8 s, respectively, according to the sensors’ specification sheets).The LED intensity was varied externally using a potentiometer, and the light intensity measurement was performed offline using a Dr Meter Luxometer (LX1330B).The output bandwidth was limited to the visible (VIS) spectrum (450 nm to 700 nm) because of the LED spectrum specifications.For mathematical modelling, known values of refractive indices (from the Filmetrics database) for Si and SiO_2_ films were used [37].

The best results, in terms of MSE and MSE_T_ values obtained from a combination of factors (in Table 2), were SENSOR2–LED2. These results are displayed in Figure 8 and Figure 9. Figure 8 shows the resulting reflectance curves for SAMPLE1 (476.3 nm) when calibrating from 85% to 95% of the maximum available counts of the ADC. Table 3 shows the RMSE and MSE_T_ results associated with Figure 8.

In Figure 8, it is observed that the measured reflectance points show a close fit to the mathematical model. Table 3 shows that all of the RMSE values were good (≤0.05), and that RMSE and MSE_T_ values could be as low as 0.022 and 0.054, respectively. This was achieved by setting the light source calibration to 85%, the potentiometer to 2.36 kΩ and the LED2 light intensity to 105.6 Lux. However, it is also seen that the MSE_T_ values were higher than the RMSE, which resulted in two warning levels when calibrating at 95% and 90%. This was expected due to the experimental nature of the device, as the MSE_T_ accounts for the standard deviation of the measured values vs the mathematical model.

Figure 9 shows the resulting reflectance curves for SAMPLE2 (198.7 nm) when calibrating from 85%, 90% and 95% of the maximum available counts of the ADC. Table 4 shows the RMSE and MSE_T_ results associated with Figure 9.

Figure 9 followed the same process as described in Figure 8. Table 4 shows similar behaviour in the RMSE, where all the measurements were <0.05. The lowest RMSE was 0.033 and its MSE_T_ was 0.848; however, the results were similar when calibrating at 95%. As expected, high warnings in the MSE_T_ were obtained. This is explained by the tilt of the measured reflectance curves, as seen in Figure 9. This tilt is mostly related to the manual alignment process to set the device in a position to enable the measurements. In the industry, reflectance systems use a fibre optic probe that directs the light to a spot on the sample; additionally, a probe stage helps to fix the probe in position to perform measurements precisely. Other interferometry-based types of equipment include x-y-z precision positioning, such as the Bruker Contour Elite, which reinforces the idea that positioning is relevant because it ensures that the interference fringes are generated correctly to perform the measurements.

The CFM helps in understanding the capabilities of the proposed reflectometry system when performing curve-fitting analysis. It shows that it is possible to match the reflectance curves corresponding to the specifications of SAMPLE1 and SAMPLE2. In addition to the CFM, the IIM can help us to confirm the thickness measurements. The following section describes thickness measurements made with the IIM.

### 4.2. Interference Interval Method (Results)

In this section, the IIM was applied as described in Section 2.4. The reflectance measurements were performed as described in Section 3.2. The following settings were used before analysing the results:Following a common practice in the industry, λ_1_ and λ_2_ were fixed to 700 nm and 450 nm, respectively, to facilitate the fringe count between the mentioned wavelength intervals (see Equation (5)). This reduces the peak/wave location complexity in Python.The IIM is an approximation method; therefore, the choice of output error is dependent on the specific application [10]. In this research, an error ≤0.10 was considered acceptable.An average of the refractive index of SiO_2_ in the VIS spectrum (*n* = 1.46) was used to calculate the thickness. The refractive index data were based on the Filmetrics database [37].Similar to the CFM results, the bandwidth under study was limited to the VIS spectrum (450 nm to 700 nm) because of the LED specifications.The IIM was not applicable for SAMPLE2 (198.7 nm) due to the known limitations of this method for SiO_2_ film thickness <400 nm.

The thickness calculation performed by SENSOR 2 is presented to describe the IIM results because this sensor presented the least variation in the CFM study (Table 2). Figure 10 shows the plotted reflectance curve and the automated thickness calculation.

Figure 10 shows the identification of one valley between λ_1_ and λ_2_, which is marked with a green dot in Figure 10; therefore the number of fringes is Δm = 1 (see Equation (5)). Considering the additional parameters stated in Figure 10, the thickness calculation was performed by the Python script. This method demonstrated a consistent measurement of 431.51 nm, compared with the SAMPLE1 thickness value of 476.3 nm. There was a difference of 44.79 nm between the measured and expected values, which represents an error of 0.09, and the results were the same for all the combinations of factors presented in this work (Table 2). The IIM results shown in Figure 10 were consistent for all the combinations of SENSOR1 and SENSOR2 in Table 2.

This section presented the thickness measurements using the developed reflectometry system. As stated, the light source quality was one of the identified opportunities whilst performing experiments. Therefore, a comparison between the best combinations of factors vs HAL/DEUT light source is presented in Section 4.3.

### 4.3. Comparison with a HAL/DEUT Light Source

The results from experiments concluded that the reflectometry setup using SENSOR2 and LED2 presented the least variation (Table 2); therefore, this configuration was selected as a baseline and compared with a HAL/DEUT light source. The materials used in this experiment are described in Section 3.1. Figure 11 shows the described setup.

The probe holder (Figure 11b) was designed in such a way that the tip of the light probe was placed at 1 mm height from the surface of the reference standards; additionally, sensor integration time was modified to 0.65 sec (int_time = 650). This was the configuration in which the HAL/DEUT light source intensity reached nearly 90% of the maximum available counts for calibration purposes. Table 5 shows a comparison between LED2 and the HAL/DEUT light source specifications.

As observed, the lamp power of the HAL/DEUT light source reduced from 78 W to 311 μW (72 + 239 μW) by using a 200 μm fibre optic probe. Additionally, the LED2 showed a reduced light emission (<0.1) in the extremes of its spectrum specifications (below 450 nm and above 700 nm); therefore, the bandwidth used for testing was 450 nm to 700 nm.

After following the measurement procedures described in this paper, the CFM and IIM were performed using the comparison setup in Figure 11. Figure 12 provides a comparative analysis of two reflectometry systems based on SENSOR2: one using LED2 and the other using the HAL/DEUT light source. The analysis is based on four different tests. Table 6 shows the data set used for the Figure 12 construction.

Figure 12 and Table 6 show that the LED2 system consistently performed better when testing with SAMPLE1. The mean values (μ) showed less errors than the HAL/DEUT light source in all the metrics. Similar results are seen with SAMPLE2, with the exception that LED2’s MSE_T_ mean was higher in this case. Therefore, the figure indicates that LED2 performed better, but there is room for improving the consistency of measurements in the reflectometry setup. The 3D-printed probe holder was designed to provide a 0° alignment for the probe, but the setup limits the repeatability of the alignment of the sensor system with the sample.

The high error values in MSE_T_ are explained by factors such as mechanical positioning, refractive index values, and sensor specifications such as spectral resolution. In future comparisons, to improve the error measurements and standard deviation further, it is important to consider other spectrometer sensors and the real refractive index per sample, which could be predicted using newer machine-learning methods [38].

On the other hand, when performing the IIM, both light sources measured an expected error of 0.09 when measuring SAMPLE1 (measured thickness: 431.51). Finally, it was not possible to measure the SAMPLE2 because of the known IIM limitations below a 400 nm thickness of SiO_2_ coating.

The use of specialised LEDs with high stability could potentially benefit the standard deviation of the measured values; furthermore, the use of LEDs would allow the scalability of spectrometry for thin film measurements in wider substrates due to their size and costs. Additionally, a single LED is cheaper by four orders of magnitude compared with a HAL/DEUT light source.

## 5. Conclusions and Future Work

A novel reflectometry system capable of measuring thin film thickness was presented in this paper. By using the curve fitting method, we have shown that it is possible to obtain a good fit with RMSE values as low as 0.022, and a normalised MSE as low as 0.054. Although there is room for improvement in the standard deviation of the measured data, the presented reflectometry system can show better results when comparing the system performance with a HAL/DEUT light source. Additionally, the presented reflectometry system was proven to work with the interference interval method, obtaining a consistent measurement error of 0.09 when measuring the thickness sample of 476.3 nm with both light sources used in this research work. The proposed system’s low cost and compact size provide a scalable option for wide-area thickness measurements of semiconductive coatings, such as SiO_2_. Compared with existing industrial measurement systems, the proposed reflectometry system offers a cost-benefit of one order of magnitude. Additionally, the developed open-source Python code will allow other researchers to replicate the experiment setup for research or academic purposes.

Future research lines derived from this work are the following: (i) the creation of an array of spectrometers for thin-film multi-point inspection in longer substrate areas, which implies the improvement of the positioning method and understanding of the limits of the presented reflectometry system for potential implementation in an R2R environment; and (ii) the combination of convolutional neural networks (CNNs) and common fitting procedures to predict thickness and refractive index measurements.

The proof of concept in this work has the potential to drive innovations in thin-film measurement sensors, fibre optic probes, wideband LEDs, and a potential novel control-feedback option for R2R processes in the future.

## Figures and Tables

**Figure 1 sensors-23-05326-f001:**
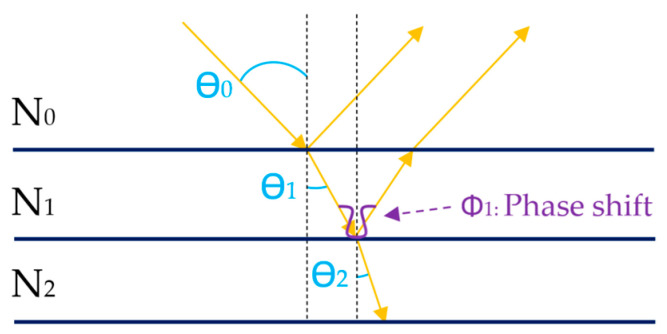
Incident light propagation of a ray of light on a thin-film coating N_1_ and a substrate N_2_. Assuming N_0_ < N_1_ < N_2_.

**Figure 2 sensors-23-05326-f002:**
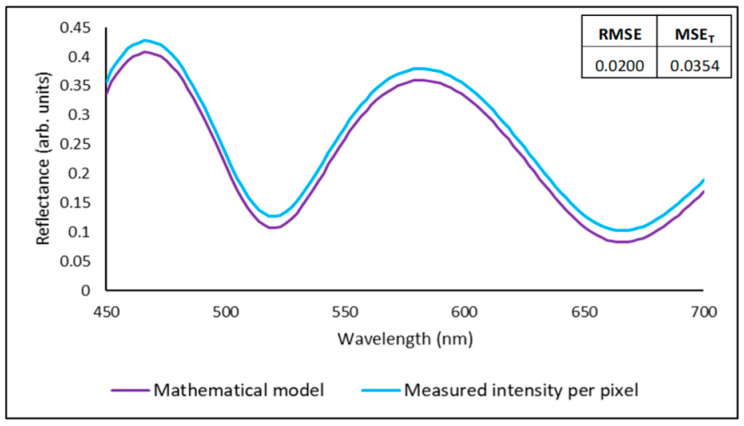
Example of the CFM showing a comparison between a modelled reflectance curve and the measured reflectance curve of a sample comprising a Si substrate and a SiO_2_ coating thickness of 800 nm.

**Figure 3 sensors-23-05326-f003:**
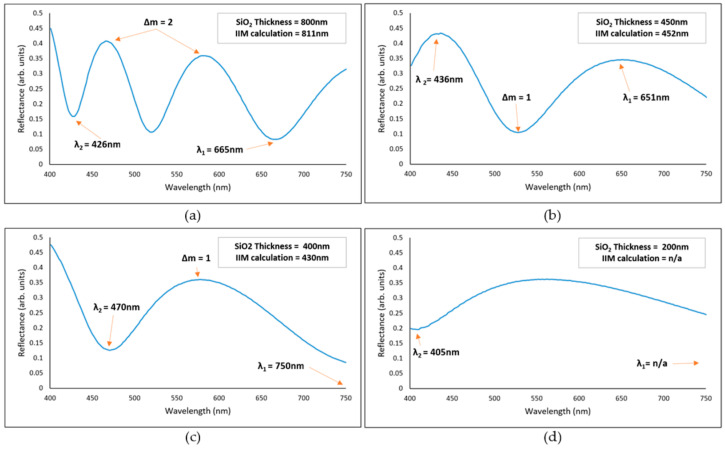
Modelled reflectance curves of different SiO_2_ thicknesses compared with the IIM thickness calculation (n = 1.46 and ϴ = 0°). (**a**) SiO_2_ thickness: 800 nm, IIM: 811 nm. (**b**) SiO_2_ thickness: 450 nm, IIM: 452 nm. (**c**) SiO_2_ thickness: 400 nm, IIM: 430 nm. (**d**) SiO_2_ thickness: 200 nm, IIM: n/a.

**Figure 4 sensors-23-05326-f004:**
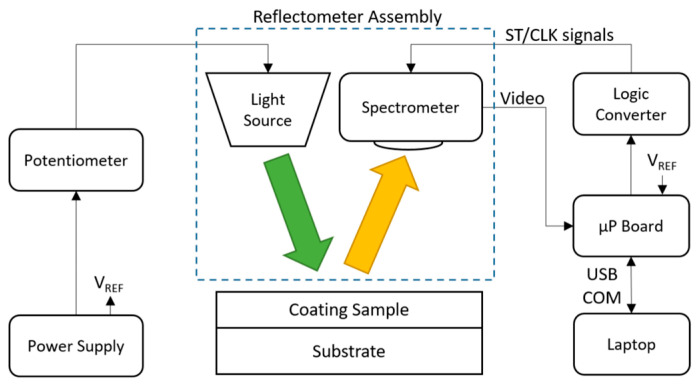
Reflectometer setup diagram.

**Figure 5 sensors-23-05326-f005:**
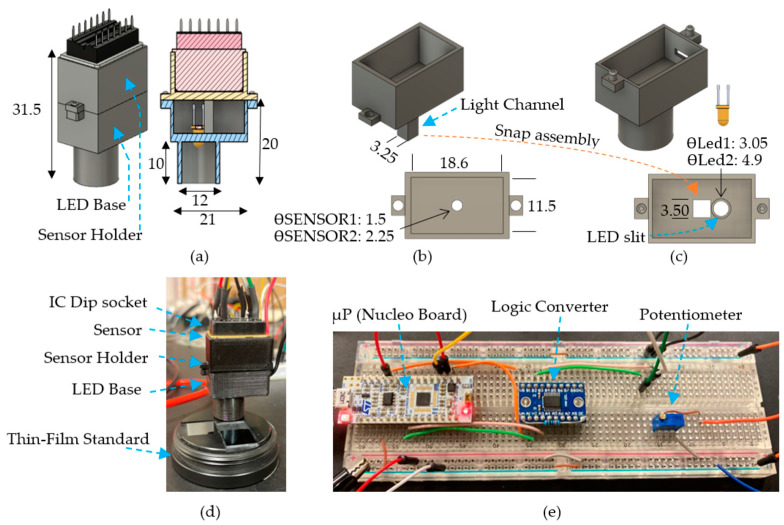
3D-printed device assembly. (**a**) 3D-printed device assembly showing a near-normal position of the LED and sensor slit and a 10 mm height from the LED to the sample. (**b**) Sensor holder Iso/top views. (**c**) LED base Iso/top views. (**d**) Physical reflectometer device and thin-film standard. (**e**) Breadboard circuit. (All dimensions in mm).

**Figure 6 sensors-23-05326-f006:**
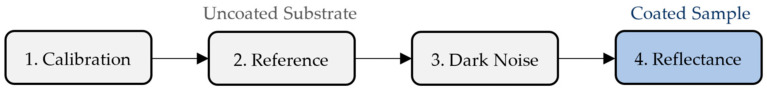
Python flow block diagram—measurement procedure.

**Figure 7 sensors-23-05326-f007:**
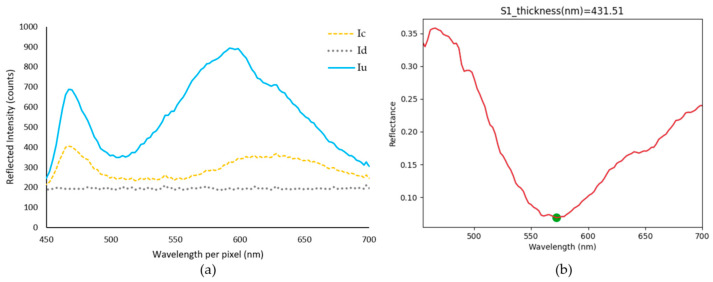
(**a**) Averaged reflected intensities of the coated sample (Ic), the uncoated sample (Iu), and the SM dark noise (Id) using LED 713-3983. (**b**) Example of a resultant Python reflectance curve showing the thickness calculation using the IIM (the green dot is a valley identified by the Python software).

**Figure 8 sensors-23-05326-f008:**
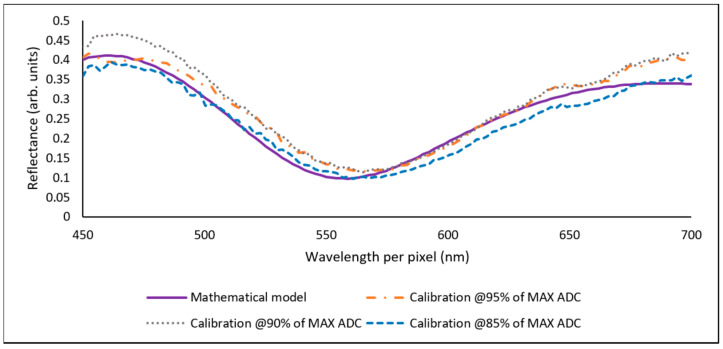
Measured reflectance data vs mathematical model. Factors: SENSOR2 (C12666MA), LED2 (713-3983) and SAMPLE1 (476.3 nm).

**Figure 9 sensors-23-05326-f009:**
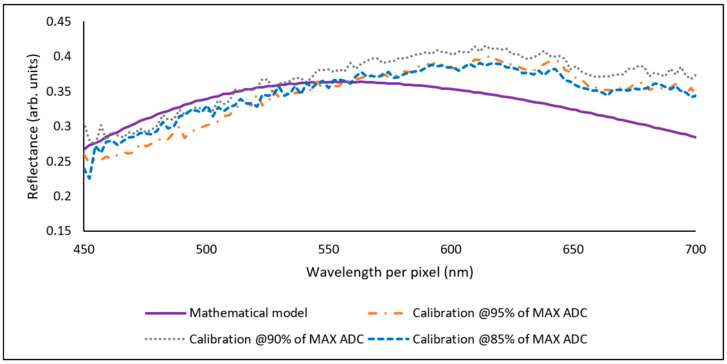
Reflectance curve vs mathematical model. Factors: SENSOR2 (C12666MA), LED2 (713-3983) and SAMPLE2 (198.7 nm).

**Figure 10 sensors-23-05326-f010:**
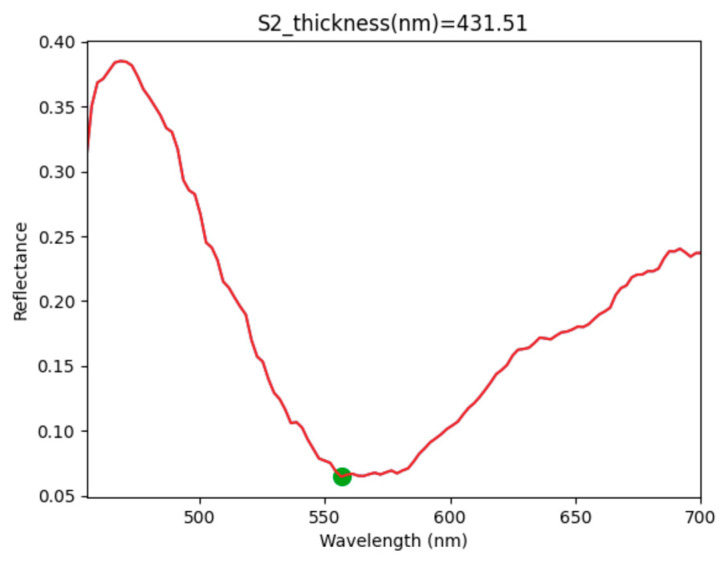
Python graph showing the IIM thickness measurement result of the SENSOR2–SAMPLE1 combination. In this case *n* = 1.46, θ = 0, λ_2_ = 450 nm, λ_1_ = 700 nm and d = 431.51 nm. The green dot is a valley identified by the Python software; therefore Δm = 1 (see Equation (5)).

**Figure 11 sensors-23-05326-f011:**
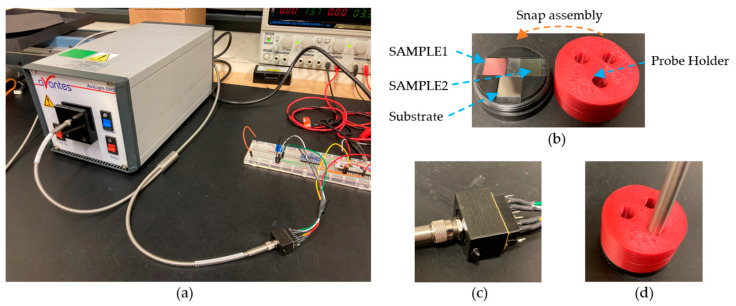
(**a**) Adapted HAL/DEUT light source in the reflectometry system using SENSOR 2 (C12666MA). (**b**) SAMPLE1 (476.3 nm), SAMPLE2 (198.7 nm), substrate (uncoated silicon) and 3D-printed probe holder. (**c**) Modified sensor holder to adapt the Thorlabs SMA adapter. (**d**) Assembly of the sensor holder light probe.

**Figure 12 sensors-23-05326-f012:**
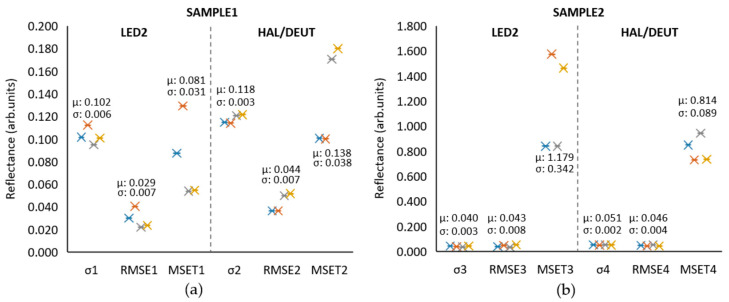
CFM comparison between LED2 and HAL/DEUT showing the standard deviation (σ), RMSE and MSET per sample and light source combination. (**a**) SAMPLE1 measurements. (**b**) SAMPLE2 measurements. (μ: mean of test points and σ: standard deviation of test points).

**Table 1 sensors-23-05326-t001:** Table of factors. * Verified with vendor measurement report (Appendix D).

	Sensors	Light Sources	Samples (SiO_2_ Thickness) *	Methods
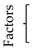	SENSOR1: C12880MA	LED1: 713-3942	SAMPLE1: 476.3 nm	IIM
SENSOR2: C12666MA	LED2: 713-3983	SAMPLE2: 198.7 nm	CFM

**Table 2 sensors-23-05326-t002:** Results. In CFM, the RMSE and MSE_T_ are considered good (green) if ≤0.05, a warning (yellow) if >0.05, and a high-level warning (orange) if >0.1. In IIM, an error is considered good if ≤0.1 or a warning if >0.1.

Experiment Factors					CFM		IIM
Exp	Method	Sensor	Led	Sample	RMSE (Lowest Value)	MSE_T_ (Norm: σ)	Error
1	CFM	SENSOR1	LED1	SAMPLE1	0.039	0.136	n/a
2				SAMPLE2	0.037	3.183	
3			LED2	SAMPLE1	0.032	0.098	
4				SAMPLE2	0.055	1.397	
5		SENSOR2	LED1	SAMPLE1	0.027	0.078	
6				SAMPLE2	0.042	1.481	
7			LED2	SAMPLE1	0.022	0.054	
8				SAMPLE2	0.033	0.84	
9	IIM	SENSOR1	LED1	SAMPLE1	n/a		0.09
10				SAMPLE2			n/a
11			LED2	SAMPLE1			0.09
12				SAMPLE2			n/a
13		SENSOR2	LED1	SAMPLE1			0.09
14				SAMPLE2			n/a
15			LED2	SAMPLE1			0.09
16				SAMPLE2			n/a

**Table 3 sensors-23-05326-t003:** Table of results for SENSOR2 (C12666MA), LED2 (713-3983) and SAMPLE1 (476.3 nm), calibrating at 85%, 90% and 95% of the MAX ADC count.

Light Source Calibration (%)	Resistor (kΩ)	Led Intensity (Lux)	RMSE	MSE_T_
95	1.38	195.2	0.030	0.087
90	1.98	134.1	0.040	0.129
85	2.36	105.6	0.022	0.054

**Table 4 sensors-23-05326-t004:** Table of results for SENSOR2 (C12666MA), LED2 (713-3983) and SAMPLE2 (198.7 nm), calibrating at 85%, 90% and 95% of the MAX ADC count.

Light Source Calibration (%)	Resistor (kΩ)	Led Intensity (Lux)	RMSE	MSE_T_
95	1.38	195.2	0.038	0.840
90	1.98	134.1	0.048	1.575
85	2.36	105.6	0.033	0.848

**Table 5 sensors-23-05326-t005:** Feature comparison between LED2 and HAL/DEUT light source. * Calculated optical power: 15 mW.

Features	Deuterium	Halogen	LED2
Wavelength range	190–400 nm	360–2500 nm	340–780 nm
Lamp power	78 W	5 W	105 mW
Noise	2 × 10^−5^	10^−4^	Not available
Colour temperature	-	3000 k	2700–3500 k
Max. drift	±0.5/hr%	±0.1/hr%	Not available
Optical power in 200 μm fibre	72 μW	239 μW	Not available
Luminous intensity	-	-	22cd *
Dimensions, weight	315 × 165 × 140 mm, 5 kg		5 × 5.3 × 28.9 mm, 0.28 g

**Table 6 sensors-23-05326-t006:** The data set of comparison between LED2 and HAL/DEUT light sources. (μ: mean of test points and σ: standard deviation of test points).

	SAMPLE1						SAMPLE2					
	LED2			HAL/DEUT			LED2			HAL/DEUT		
Test	σ_1_	RMSE_1_	MSE_T1_	σ_2_	RMSE_2_	MSE_T2_	σ_3_	RMSE_3_	MSE_T3_	σ_4_	RMSE_4_	MSE_T4_
1	0.102	0.030	0.087	0.115	0.036	0.101	0.042	0.038	0.840	0.053	0.049	0.849
2	0.112	0.040	0.129	0.114	0.036	0.100	0.038	0.048	1.575	0.049	0.042	0.730
3	0.095	0.022	0.054	0.121	0.050	0.171	0.035	0.033	0.840	0.053	0.052	0.944
4	0.101	0.024	0.055	0.122	0.052	0.180	0.043	0.053	1.463	0.051	0.043	0.734
μ	0.102	0.029	0.081	0.118	0.044	0.138	0.040	0.043	1.179	0.051	0.046	0.814
σ	0.006	0.007	0.031	0.003	0.007	0.038	0.003	0.008	0.342	0.002	0.004	0.089

## Data Availability

The dataset supporting this study is openly available in: (Under approval) https://doi.org/10.15131/shef.data.23285603.

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
