# Peer review of "A Spectroscopic Reflectance-Based Low-Cost Thickness Measurement System for Thin Films: Development and Testing"

_sensors, 2023, doi:10.3390/s23115326_

Round 1
Reviewer 1 Report
The paper “A spectroscopic reflectance-based low-cost thickness measurement system for thin films: development and testing” presents the hardware and software development of a novel low-cost spectroscopic reflectance system using two state-of-the-art sensors for thin-film thickness measurements. There are some doubts:
1. Comparing the refractive indices of N1 and N2 in part of Figure 1, if N1 is larger than N2, the yellow light in N2 of the original figure is correct. If N1 is smaller than N2, then θ1 should be larger than θ2.
2. Good fits with RMSE values as low as 0.022 and normalized MSE as low as 0.054. The sample used was measured as many times as the standard deviation of the measured values is mentioned in the text. The standard deviation of the single measurements is not convincing enough.
Minor editing of English language required.
Reviewer 2 Report
The authors presented a low-cost spectroscopic reflectance system for the measurement of film thickness based on two methods including the curve fitting method (CFM) and the interference interval method (IIM). Root mean square error (RMSE) and normalized mean square error (MSE) were used to evaluate the reliability of the proposed system, in which the best combination of components reached an RMSE of 0.022 and an MSE of 0.054 for the curve fitting method. Additionally, the interference interval method also showed an error of 0.09 when compared to the modeled values. The low-cost yet effective system for the reflectance measurement of thin films can be used as an alternative in the roll-to-roll process to meet the rapid growth of the manufacturing of flexible electronics. Below are some comments to further improve the results:
1. The author discussed the limitation of the interference interval method where at least two peaks or through are needed within the measured wavelength to calculate the film thickness. This is the reason why IIM was not applied to the sample with a SiO2 thickness of 100 nm. Is there a way to circumvent this problem by extending the range of measurement or introducing interference by coating?
2. Both CFM and IIM assume the refractive index of the thin film is a known value. However, the reality the refractive index is not constant for different wavelengths and may change from sample to sample. Deep learning models such as neural networks have been used to extract information about the refractive index from the reflection measurement. (2D Mater. 10, 025025, 2023. DOI 10.1088/2053-1583/acc59b) Such analysis would not be limited by the thickness of the film (whether it supports interference) or the unknown refractive index of new materials. Do the authors have a specific plan to incorporate advanced machine-learning models into their system?
